# Calculating human thermal comfort and thermal stress in the PALM model system 6.0

Fröhlich Dominik[1] and Matzarakis Andreas[1]

[1]Deutscher Wetterdienst (DWD), Research Centre Human Biometeorology, Stefan-Meier-Str. 4, 79104 Freiburg

**Correspondence:** Dominik Fröhlich (dominik.froehlich@mailbox.org)

**Abstract.** In the frame of the project "MOSAIK – Model–based city planning and application in climate change", a German–wide research project within the call "Urban Climate Under Change" ($[UC]^2$) funded by the German Federal Ministry of Education and Research (BMBF), a biometeorology module was implemented into the PALM model system. The new biometeorology module comprises of methods for the calculation of uv-exposure quantities, a human–biometeorologically weighted mean radiant temperature ($T_{mrt}$), as well as for the estimation of human thermal comfort or stress. The latter is achieved through the implementation of the three widely–used thermal indices Perceived Temperature (PT), Universal Thermal Climate Index (UTCI), as well as Physiologically Equivalent Temperature (PET) . Comparison calculations were performed for the indices PT, UTCI and PET based on the SkyHelios model and showing PALM calculates higher values in general. This is mostly due to a higher radiational gain leading to higher values of mean radiant temperature. For a more direct comparison, the indices PT, PET and UTCI were calculated by the biometeorology module, as well as the programs provided by the attachment to the VDI guideline 3787, as well as by the RayMan model based on the very same input dataset. Results show deviations below the relevant precision of 0.1 K for PET and UTCI and some deviations of up to 2.683 K for PT caused by repeated unvaovourable rounding in very rare cases (0.027 %).

## 1 Introduction

Urban areas show slightly different diurnal variability in air temperature ($T_a$) compared to their surroundings (e.g. Oke, 1995; Helbig et al., 1999). This is mostly due to modifications in the radiation budget caused by ground sealing, different surface materials and many vertical surfaces (Oke, 1995, p. 276ff). Additionally many of them have high heat storage capacities (Oke, 1995, p. 284) reducing night–time cooling. The two effects contribute to a phenomenon that is called the Urban Heat Island (UHI, Oke, 1995, p. 288ff). Another increase in urban temperatures is caused by the local impact of global climate change. E.g. for Freiburg (south–west Germany), an increase of days with heat stress by up to 5 % is expected (Matzarakis and Endler, 2010).

Health and well–being of the growing urban population is already an important issue in present urban planning (e.g. Helbig

et al., 1999). A number of studies have been carried out in the last years that show strong correlation between health, as well as mortality on the one side and urban biometeorology on the other side. Especially heat stress during the summer months seems to lead to an increase in mortality (e.g. Koppe et al., 2004; Conti et al., 2005; Muthers et al., 2010; Nastos and Matzarakis, 2012; Muthers et al., 2017).

5 To allow for counteracting malicious effects through urban planning meassures, e.g. by a modification in the building config- uration (Lin et al., 2010a), surface materials (Lin et al., 2010b) or urban green (Shashua-Bar et al., 2011; Charalampopoulos et al., 2015) decision makers are dependent on spatially resolved thermal perception information that can be best provided through maps (Matzarakis, 2001; Nouri et al., 2018).

Thermal comfort can be assessed by calculating thermal indices, e.g. the Predicted Mean Vote (PMV Fanger, 1972), Physiolog- 10 ically Equivalent Temperature (PET, Höppe, 1993, 1999), the Perceived Temperature (PT, Staiger et al., 2012) or the Universal Thermal Climate Index (UTCI, Jendritzky et al., 2012) combining several aspects to approximate the thermal perception of a standardized sample human being taking into account many meteorological and physiological parameters (Fanger, 1972; Höppe, 1999; Staiger et al., 2012, 2019).

To facilitate the identification of hotspots and the assessment of potential for the reduction of thermal stress the program "Urban 15 Climate Under Change" ($[UC]^2$) is funded by the German Federal Ministry of Education and Research (BMBF). It "aims at the development, validation and application of an innovative urban climate model for entire cities" (Todo: UC2 homepage). Part of the $[UC]^2$ program is the German-wide research project "MOSAIK – Model–based city planning and application in climate change". In the course of MOSAIK the PALM (PArallelized Large–eddy simulation Model Raasch and Schröter, 2001; Maronga et al., 2015; Hellsten et al., 2018; Maronga et al., 2019b) is extended by several modules to extend it to become a 20 comprehensive urban climate model (e.g. an urban surface module by Resler et al. (2017)). One of the new modules is the biometeorology module capable of calculating the static thermal indices PT, UTCI and PET (Maronga et al., 2019b).

## 2 Methods

Humans are unable to directly sense individual meteorological quantities, e.g. $T_a$. However, they do feel the thermal effect of their environment caused by several meteorological parameters integrally through the skin and the blood temperature in the 25 thermoregulatory system of the hypothalamus (Tromp, 1980; Höppe, 1993). Thermal comfort therefore can not be described by individual parameters, but needs to be approximated through thermal comfort indices considering all relevant conditions. The more sophisticated indices are based on the approach of equivalent temperaturs and are relying on the evaluation of the human energy balance or heat flux models (e.g. Fanger, 1972; Gagge et al., 1986; Höppe, 1993; Błażejczyk et al., 2012).

From the meteorological side, comprehensive thermal indices usually do require input for the meteorological parameters air 30 temperature ($T_a$), vapor pressure (VP), wind speed (WS) and the mean radiant temperature ($T_{mrt}$), defined as the temperature of a perfectly black environment causing thermal radiation only, that leads to the same radiational gain or loss than the actual environment (Fanger, 1972; Thorsson et al., 2007). All input conditions are required at the very loaction the index is calculated for in a height of 1.1 m representing the gravimetric center of an average human body (Fanger, 1972). Due to the discrete

design of PALM, the biometeorology module can only obtain information at cell centers. It therefore calculates thermal indices for the vertical cell level with the height of the cell center closest to 1.1 m above ground level.

The estimation of $T_{mrt}$ does require radiational input data, that is provided by one of the two radiation schemes available in PALM, the simple clear-sky model (Maronga et al., 2019a), or the more complex Rapid Radiative Transfer Model (RRTM)
(Mlawer et al., 1997; Pincus et al., 2003; Clough et al., 2005; Maronga et al., 2019a).

The simple clear-sky model generates radiation data based on astronomic calculations, namely the solar constant assumed as 1368 $\frac{W}{m^2}$ as well as the losses by the transmissivity of the atmosphere estimated through geometrical calculations of the solar position only. The impact of clounds, moisture and atmospheric variations is ignored (please refer to Maronga et al., 2019a, sec 3.6.1).

The more complex and more precise estimation of radiation data is derived from incorporating the RRTMG model. It does e.g. allow for the consideration of clouds and other weather effects (Mlawer et al., 1997). However, both models do only provide radiation fluxes at one energy transfer surface and are therefore insufficient for the estimation of the mean radiant temperature in urban areas.

For complex environments, PALM-4U 6.0 contains the Radiative Transfer Module, that is driven by the clear-sky model or
RRTMG at the upper model border to estimate radiation fluxes within the canopy layer. It incorporates a Building Surface Model and a Land Surface Model to consider the effects of buildings and vegetation on the short- and longwave radiation fluxes at individual grid cells. (please refer to Maronga et al., 2019a, sec 4.4).

## 2.1 Perceived Temperature

The Perceived Temperature (PT) is a thermal comfort index for outdoor environments using the concept of an equivalent temperature. The thermal impact of the environment is evaluated through the "Klima–Michel–Model" (Jendritzky et al., 1990), an energy balance model for human beings (Staiger et al., 2012). PT is defined to be "the air temperature of a reference environment in which the thermal perception would be the same as in the actual environment" (Staiger et al., 2012).

PT is a steady–state model by design to keep run–time at a reasonable level. The target for PT is a standardized sample human
(the "Klima–Michel", Jendritzky et al., 1990) with a height of 1,75 m, an age of 35 years, a weight of 75 kg, an internal heat production of 135 W/m² walking at a speed of 4 km/h (Staiger et al., 2012). This allows for a simplification of the human heat balance equation after ASHRAE (2001, p. 134):

$$M - Wo = (C + R + E_{sk}) + (C_{res} + E_{res}) + S_{sk} + S_{cr} \tag{1}$$

The energy gain caused by metabolic processes within the body $M$ reduced by the portion of mechanical work $Wo$ (the frac-
tion of the body's energy, that is not converted to heat, but to mechanical workforce) is compared to the combined latent and sensible heat fluxes from or to the environment. The components of the equation represent energy transfer by sensible heat $C$, radiation $R$, and latent heat $E$. Eq. 1 distinguishes between fluxes from or to the skin ($_{sk}$), the core ($_{cr}$) and through the respiratory system ($_{res}$). The heat storage components ($S$) are considered to equal 0 W constantly assuming a steady–state.

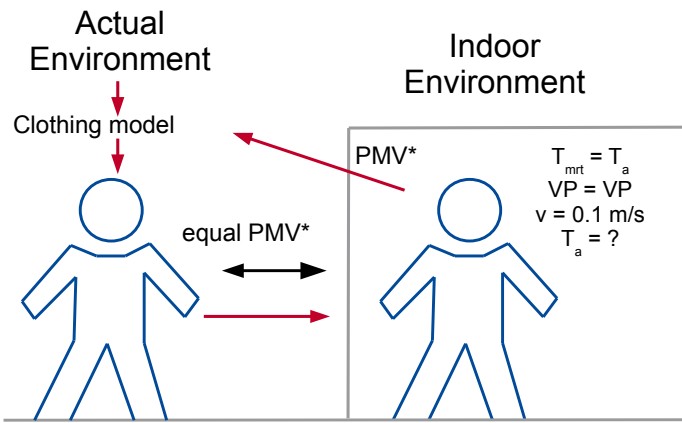

**Figure 1.** Schematic overview of the comparison of adjusted PMV between the actual prevailing environment and a virtual eindoor environment for the estimation of the perceived temperature. The sample human is standardized by the "Klima–Michel" model.

Unit of all parameters is W.

All of the physiological parameters are defined by the "Klima–Michel" model and the clothing model is self-adapting. PT can therefore be estimated exclusively based on the meteorological parameters air temperature ($T_a$, °C), wind speed(v, m/s), vapor pressure (VP, hPa), and mean radiant temperature ($T_{mrt}$, °C). All of the energy gained or lost by the "Klima-Michel"

is compared to that of an "indoor" reference environment (compare to Figure 1). This is done based on a modified version of the basic thermal index "Predicted Mean Vote" (PMV) after Fanger (1972); Gagge et al. (1986). The reference environment is defined with parameters $T_{mrt} = T_a$ (no radiational impact), v = 0.1 m/s (auto-convection only) and VP equal to VP of the actual environment. If the actual environment would lead to warm and humid conditions, VP is set to a value matching a relative humidity of 50 % (Staiger et al., 2012). The comparison is balanced by the air temperature of the "indoor" environment that is

modified until the thermal stress in terms of PMV is the same as in the actual environment.

The index PMV does consider energy exchange based on a two-node body model (a skin and a core node). It allows for latent and sensible heat transfer from or to the skin (considering sweating) and by respiration (Fanger, 1972; Staiger et al., 2012).

PT comprises a clothing model, that is automatically selecting the most appropriate value for the clothing index (clo) according to the prevailing meteorological conditions (Staiger et al., 2012). It primarily attempts to maintain thermal comfort by adapting

to hot or cold conditions. Only if this can not be achieved, thermal stress is computed (Fanger, 1972; Staiger et al., 2012). The clothing model is supported in reducing thermal strain by parametrizations of shivering in cold conditions (PMV < -0.11 at clo = 1.75) and sweating under hot conditions (PMV > 0.5 at clo = 0.5, Staiger et al., 2012).

To facilitate the interpretation of PT results in Central Europe Staiger et al. (2012) published a perception table translating the

PT values into thermal perception or the extent of thermo-physiological stress (Table 1).

**Table 1.** The thermo-physiological meaning of PT results for central Europe as defined by Staiger et al. (2012).

| PT ($^{\circ}$C) | Thermal Perception | Thermo-physiological stress |
| --- | --- | --- |
| $\geq$ +38 | Very hot | Extreme heat stress |
| +32 – +38 | Hot | Great heat stress |
| +26 – +32 | Warm | Moderate heat stress |
| +20 – +26 | Slightly warm | Slight heat stress |
| 0 – +20 | Comfortable | Comfort possible |
| -13 – 0 | Slightly cool | Slight cold stress |
| -26 – -13 | Cool | Moderate cold stress |
| -39 – -26 | Cold | Great cold stress |
| < -39 | Very cold | Extreme cold stress |

### 2.1.1 Universal Thermal Climate Index

The Universal Thermal Climate Index (UTCI) is "the isothermal air temperature of the reference condition that would elicit in the same dynamic response (strain) of the physiological model" than the actual environment Jendritzky et al. (2012)

Alike most complex thermal indices (e.g. PT or PET), UTCI is an equivalent temperature. The thermal effect of the prevailing meteorological conditions is compared to the one of a standardized reference "indoor" environment with a fix 50 % relative humidity, calm air (0.1 m/s) and $T_{mrt}$ equal to $T_a$ (Jendritzky et al., 2012). The environments are compared by a heat transfer model introduced by Fiala et al. (2012).

For performance reasons, UTCI can only be approximated using a regression equation abbreviated from sample calculations performed by computing centers (Jendritzky et al., 2012; Bröde et al., 2012). It allows for a computationally cheap and highly performant determination of UTCI. However, it also causes a limited range of input parameters it can deal with. The regression equation supports $T_a$ in the range of -50.0$^{\circ}$C to +50.0$^{\circ}$C, a relative humidity from 0 % to 100 %, wind speed of at least 0.5 m/s and up to 17.0 m/s, as well as a difference between $T_{mrt}$ and $T_a$ ($T_{mrt}$ - $T_a$) of -30.0$^{\circ}$C to +70.0 $^{\circ}$C. In case the local meteorological conditions are out of bounds, specific workarounds after Bröde et al. (2012) are implemented.

In contrast to other indices, UTCI is determined based on wind speed in 10 m above ground level instead of 1.1 m (Bröde et al., 2012). This is for UTCI was designed for the use with meteorological station data as well as weather prediciton models, that usually provide wind speed at that level (Jendritzky et al., 2012). For the use with the biometeorology module, however, this would cause imprecision for the wind speed at 10 m above ground level is hardly representative at street level. Furthermore, wind speed for 1.1 m height is determined within the UTCI calculations based on a power-law profile (Havenith et al., 2012), that can hardly be assumed valid within the urban canopy. For the profile is part of the regression equation and can not be removed, the biometeorology module does apply the very same profile to extrapolate wind spped at 1.1 m height to obtain the input wind speed in 10 m height. This removes imprecision caused by the invalid profile as well as issues arising from obstacles

above the target height (e.g. bridges or trees).

Due to the evaluation by the regression equation, physiological parameters can not be modified in UTCI and are considered to be static. UTCI does assume a permanent walking speed of 4 km/h (1.11 m/s) resulting in an internal heat production of 135 W/m² (Jendritzky et al., 2012) and the clothing insulation to be self-adapting according to the environmental conditions (Havenith et al., 2012).

As long as all input conditions are in range for the regression equation UTCI is quite sensitive to wind speed (Chen and Matzarakis, 2018; Fröhlich and Matzarakis, 2016), but also to $T_a$ and $T_{mrt}$ (Chen and Matzarakis, 2018; Fröhlich and Matzarakis, 2016).

**Table 2.** Thermal stress classification for UTCI. Modified after Błażejczyk et al. (2013).

| UTCI (°C) | Thermal Stress category |
|---|---|
| $\geq$ +46 | Extreme heat stress |
| +38 – +46 | Very strong heat stress |
| +32 – +38 | Strong heat stress |
| +26 – +32 | Moderate heat stress |
| +9 – +26 | No thermal stress |
| 0 – +9 | Slight cold stress |
| -13 – 0 | Moderate cold stress |
| -27 – -13 | Strong cold stress |
| -40 – -27 | Very strong cold stress |
| < -40 | Extreme cold stress |

Błażejczyk et al. (2013) published a thermal stress classification for Central Europe allowing for the interpretation of UTCI results (2). In contrast to the assessment tables for PT (1) and PET (e.g. 3), the UTCI assessment table is a thermal stress classification (Błażejczyk et al., 2013) rather than a thermal comfort evaluation.

### 2.1.2 Physiologically Equivalent Temperature

The Physiological Equivalent Temperature (PET) can be considered to be one of the most popular thermal index and is widely used for the assessment of human thermal comfort. Höppe (1999) defiens PET as "the air temperature at which, in a typical indoor setting (without wind and solar radiation), the energy budget of the human body is balanced with the same core and skin temperature as under the complex outdoor conditions to be assessed" (Mayer and Höppe, 1987; Höppe, 1999, compare to Figure 2). PET evaluates heat load based on a simplified human energy balance model by (Höppe, 1984), the "Munich Energy Balance Model for Individuals" (MEMI, Höppe, 1984). PET does not comprise a self-adapting clothing model, but is entirely depending on the user input. It therefore does not include any behavioural components making PET "a real climatic index

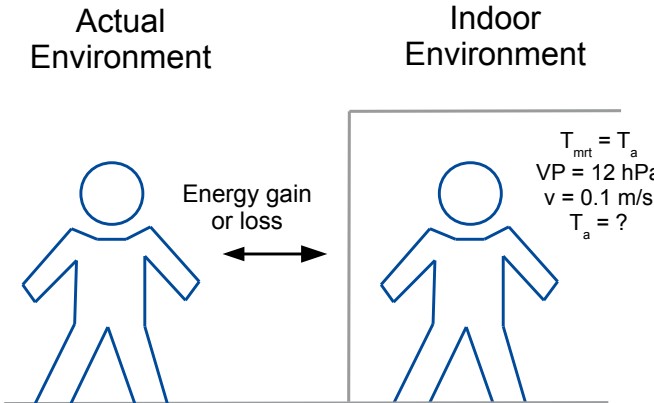

**Figure 2.** Schematic overview of the comparison of the energy gain or loss in the heat ballance equation between the actual prevailing environment and a virtual eindoor environment for the estimation of the physiologically equivalent temperature. The sample human is represented by the MEMI model.

describing the thermal environment in a thermo-physiologically weighted way" (Höppe, 1999).

PET is very sensitive to the input parameter $T_{mrt}$ (°C, Charalampopoulos et al., 2013; Chen and Matzarakis, 2018). It does also respond strongly to modifications in wind speed (v) and $T_a$ (°C). Air humidity in terms od vapour pressure (hPa)must be provided as input, but only shows very weak impact on PET (e.g. Chen and Matzarakis, 2018; Fröhlich and Matzarakis, 2016).

The thermal environment is evaluated by the human energy balance equation (2, Höppe, 1999).

$$M + Wo + R + C + E_{sk} + E_{res} + E_{sw} + S = 0 \tag{2}$$

It does consider the metabolic heat production ($M$), the mechanical workload ($Wo$), radiational heat flux ($R$), sensible heat flux ($C$), as well as latent heat ($E$). $E$ is thereby separated in the components from or to the skin ($_{sk}$), by sweating ($_{sw}$) and by the respiratory system ($_{res}$). The unit of all components of equation 2 is $W$. Heat storage ($S$) must permanently equal 0 $W$ to

maintain a steady state.

The energy gain or loss by the prevailing thermal environment is compared to that of an virtual "indoor" environment without radiational impact ($T_{mrt} = T_a$), calm air (v = 0.1 m/s), and static humidity in terms of VP = 12 hPa (Höppe, 1999). $T_a$ of the indoor environment is then modified until the indoor environment is causing the same thermal load than the actual environment. The $T_a$ of that indoor environment then is returned as PET (Höppe, 1999).

PET results can be interpreted using classification tables for the region in question. For Central Europe a classification with nine classes of thermal perception (3) was introduced by Matzarakis and Mayer (1996).

**Table 3.** Thermal sensation classes for human beings in Central Europe (with an internal heat production of 80 W and a heat transfer resistance of the clothing of 0.9 clo (clothing value)) modified after Matzarakis and Mayer (1996).

| PET (°C) | Thermal Perception | Grade of physical stress |
|----------|--------------------|--------------------------|
| > 41     | Very hot           | Extreme heat stress      |
| 35 – 41  | Hot                | Strong heat stress       |
| 29 – 35  | Warm               | Moderate heat stress     |
| 23 – 29  | Slightly warm      | Slight heat stress       |
| 18 – 23  | Comfortable        | No thermal stress        |
| 13 – 18  | Slightly cool      | Slight cold stress       |
| 8 – 13   | Cool               | Moderate cold stress     |
| 4 – 8    | Cold               | Strong cold stress       |
| ≤ 4      | Very cold          | Extreme cold stress      |

## 2.2 Test case

The thermal comfort part of the biometeorology module was tested based on the generic urban crossroads test–case "test_urban" located at Hannover (Germany) (Fröhlich, 2019, compare to Figure 3). It consists of a 19 x 19 x 60 grid domain with a grid spacing of 2.0 x 2.0 x 2.0 m. In the corners of the domain there are buildings with different heights of 10 m to 40 m. Two streets in between are forming a crossroads. In the North-East of the domain shading is provided by two deciduous trees.

Radiation data for the test case is generated by PALMs clear-sky schmeme providing minimal radiation input based on astronomic calculations assuming a perfectly clear sky without any clouds or obstructions. Please see Section 3.5.1 in Maronga et al. (2019a) for details.

To run the test setup with the thermal comfort part of the biometeorology module, the input file "test_urban_p3d" was slightly modified (Fröhlich, 2019, please see "test_urban_v2.zip/INPUT/test_urban_p3d"). The date was set to the 6[th] of March to obtain less extreme conditions. The initial potential temperature was adjusted respectively to better meet typical conditions in March. It was set to 5.0 °C at the surface at startup. The meteorological initial conditions can be found in table **??**.

For the assessment of the quality of the results, comparison calculations were performed for 07:00 UTC and 13:00 UTC of a 6[th] of March using the well-known and frequently applied SkyHelios model (Fröhlich and Matzarakis, 2018; Fröhlich, 2017; Matzarakis and Matuschek, 2011). Therefore a similar test domain was created for the SkyHelios model (see Figure 3). To increase comparability, the test calculations were driven by the average air temperature calculated by PALM.

## 2.3 Meteorological Data

For a direct comparison based on the very same input, the thermal indices provided by the biometeorology module were calculated for a meteorological dataset recorded by a urban climate station on top of the chemistry highrise building of the

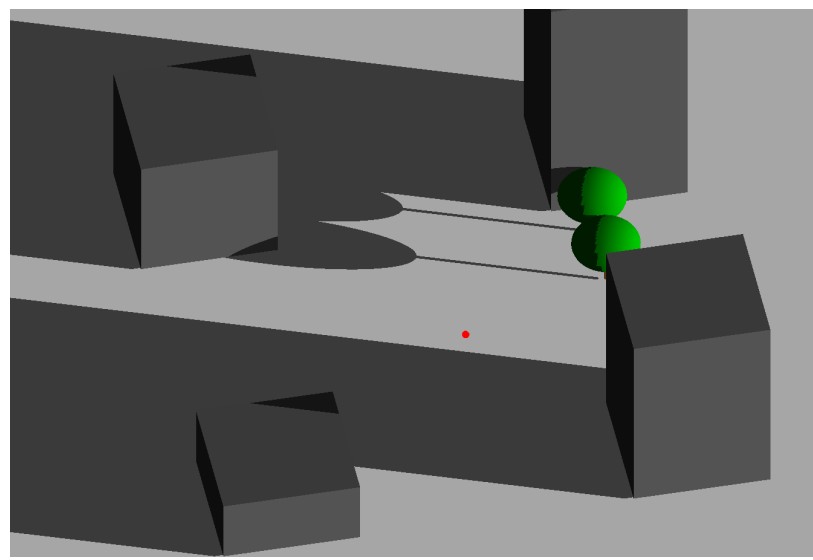

**Figure 3.** The site "test_urban" with shading as shown by the SkyHelios model seen from South-SouthWest at 07:00 UTC (shortly after sunrise) on a $6^{\text{th}}$ of March.

**Table 4.** Overviewv over the meteorological initial conditions to run the test cases in PALM.

| | |
|---|---|
| Date | $6^{\text{th}}$ of March |
| Time | 00:00:00 (UTC+1) |
| Air temperature | 5.0 °C |
| Surface water vapor mixing ratio | 0.001 (kg/kg) |
| Wind speed | 1.0 m/s |
| Wind direction | 270° |
| Cloud cover | 0 / 8 |

University of Freiburg. The dataset does cover the timespan from 1999-09-01 00:00 LST to 2010-04-30 23:00 LST in 10 minutes resolution and provides the parameters $T_a$, VP, v and global radiation, that was used to estimate $T_{mrt}$ by the RayMan model. The general statistics of the dataset is provided by Table 5. The output generated by the biometeorology module was then compared to the output by the programs in the attachment to the VDI guideline 3787, part II (VDI, 2008) and to the output
5  by the RayMan model (Matzarakis et al., 2007, 2010).

## 3   Results

For keeping the manuscript at a reasonable size only two exampleas are presented here. However, the entire dataset with input and output is published along with the manuscript and can be found at https://zenodo.org/record/3433720.

**Table 5.** Statistical overview over the meteorological data applied in the comparison of the thermal indices implemented in the biometeorology module to the reference implementations provided by the VDI guideline 3787, part II (VDI, 2008).

|  | $T_a$ | VP | v | $T_{mrt}$ |
|---|---|---|---|---|
| Min. | -13.8 | -0.8 | 0.1 | -26.0 |
| 1st Qu. | 6.7 | 6.3 | 0.4 | 0.4 |
| Median | 12.7 | 9.1 | 0.7 | 8.7 |
| Mean | 12.6 | 9.7 | 1.0 | 11.9 |
| 3rd Qu. | 18.4 | 12.5 | 1.3 | 19.7 |
| Max. | 40.1 | 43.6 | 6.2 | 64.8 |
| NA's | 279 | 2034 | 15291 | 17073 |

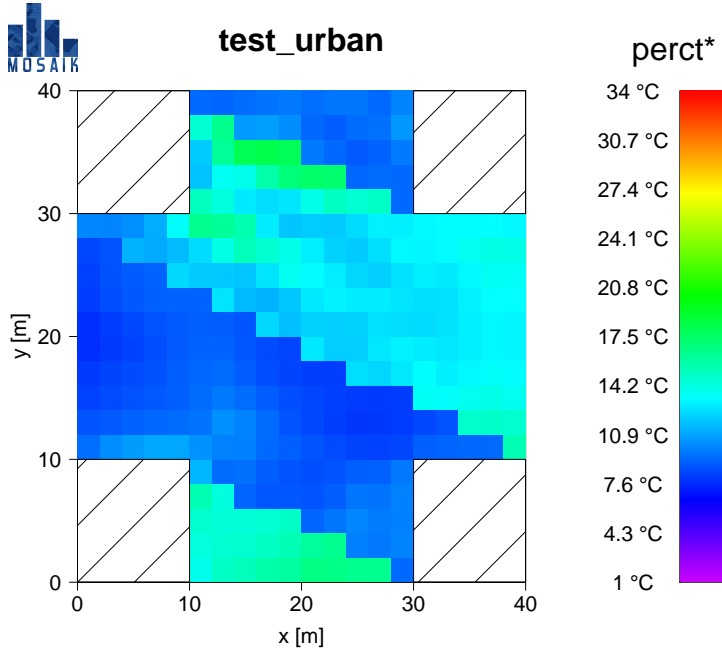

bio_perct*_xy [°C] after 25200.592 s / 07:00 UTC

**Figure 4.** Perceived Temperature (PT, perct* in palm) for the test case "test_urban" at 07:00 UTC (shortly after sunrise) on a 6[th] of March. Incident wind is from 270° with 1.0 m/s.

Looking at the perceived temperature (Figure 4) the day starts quite warm with PT of 10.2 to 15.6 °C in the sun and 8.0 to 11.1 °C in shaded areas shortly after sunrise at 07:00 UTC. The differences within the shaded or sunny areas thereby are mostly caused by wind speed. The longwave emissions of the walls, even if they are exposed to direct radiation, what can

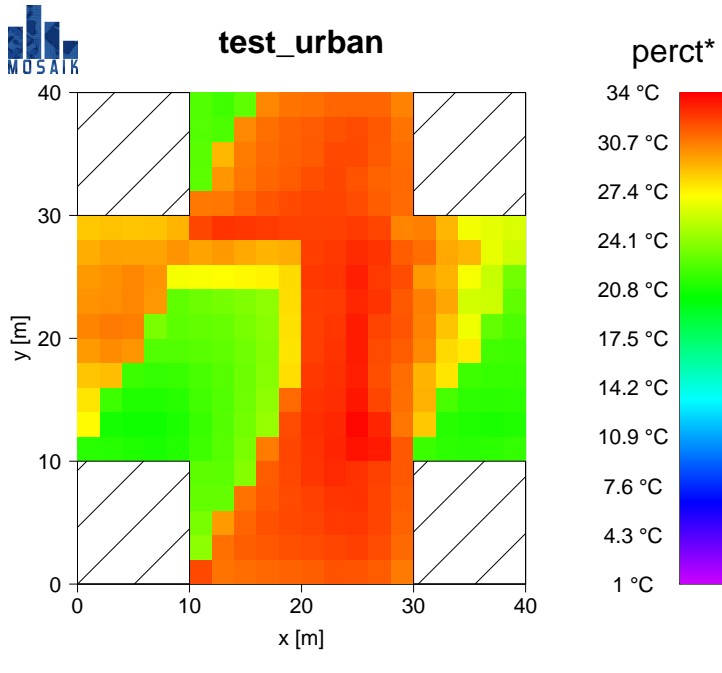

bio_perct*_xy [°C] after 46800.678 s / 13:00 UTC

**Figure 5.** Perceived Temperature (PT, perct* in palm) for the test case "test_urban" at 13:00 UTC (close to midday) on a 6[th] of March. Incident wind is from 270° with 1.0 m/s.

nicely be seen on the eastern side of the building in the lower left of Figure 4, are weak for surface temperatures are lower than the air temperature. The warm conditions for early spring are caused by a relatively high air temperature of 10.3 - 12.2 °C, that also lead to quite high mean radiant temperature of 11.4 - 14.5 °C in shaded areas and 20.1 - 35.4 °C in the sun. Wind speed is rather low throughout the model domain ranging from less than 0.1 m/s to 0.5 m/s.

The thermo-physiological consequences for a sample human, passing through the model domain are indifferent. According to the thermal perception table for Central Europe, Table 1, all readings are within the class 0 to 20 °C and, thus, can be perceived as comfortable if appropriate clothing is selected. This holds for both, shaded areas, as well as areas exposed to direct radiation.

The same scenario looks entirely different after midday at 13:00 UTC (see Figure 5). The model's "clear-sky" radiation scheme causes the air temperature to rise to values of 20.8 °C close to the northern wall of the lower right building to 24.3 °C

at the western side of the lower right obstacle. Wind speed is little decreased compared to 07:00 and ranges from less than 0.1 m/s to 0.4 m/s at 13:00. Both leads to a quite high mean radiant temperature of 25.7 - 32.7 °C in shaded areas and a very high $T_{mrt}$ of 44 - 51.7 °C in areas exposed to direct radiation.

A sample human roaming within the model domain would experience wider range of thermal perception. While shaded areas are quite comfortable with PT of 20.0 - 23.4 °C, what translates to "slightly warm" perception according to Table 1, the high

$T_{mrt}$ in unshaded areas also cause high values for PT of 24.4 - 30.9 °C. According to the thermo-physiological perception

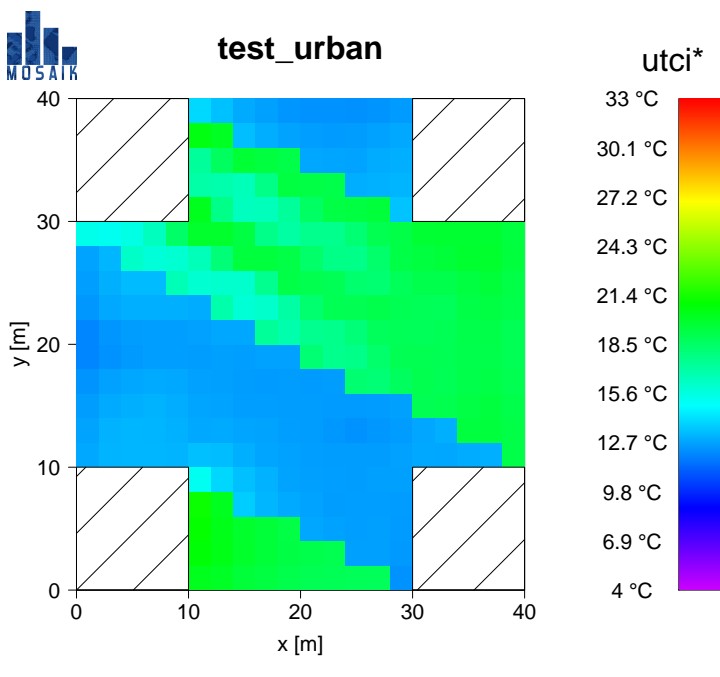

bio_utci*_xy [°C] after 25200.592 s / 07:00 UTC

**Figure 6.** Universal Thermal Climate Index (UTCI, utci* in palm) for the test case "test_urban" at 07:00 UTC (shortly after sunrise) on a 6[th] of March.

classification by Staiger et al. (2012, Table 1), the sample person would experience "slightly warm" to "warm" conditions causing slight to moderate heat stress.

The same scenario can also be analyzed targeting thermal stress using the thermal index UTCI (see Section 2.1.1). For 07:00 UTCI calculates quite similar values than PT (compare Figures 4 and 6). The absolute numbers for UTCI are way higher than those for PT with 11.5 - 14.4 °C (UTCI) in the shade and 15.7 - 19.2 °C (UTCI) in sunlit areas. This, however translates to comfortable conditions without thermal stress throughout the entire model domain (compare to Table 2) and therefore is in good agreement with the results for PT.

Taking a closer look at Figure 6 one can see, that the results for UTCI appear to be more homogeneous in some areas than those for PT (compare to 4). One of those areas can be found in between the buildings on the right with UTCI of 18.1 - 18.3 °C. They are mostly caused by wind speed going below the valid range for wind speed to the UTCI regression equation (see Section 2.1.1).

### 3.1 Comparison with SkyHelios

A similar model domain was created for the SkyHelios model (Fröhlich and Matzarakis, 2018; Fröhlich, 2017; Matzarakis and Matuschek, 2011) and a run with similar input parameters was performed. Results for the Perceived Temperature (see Figure 7)

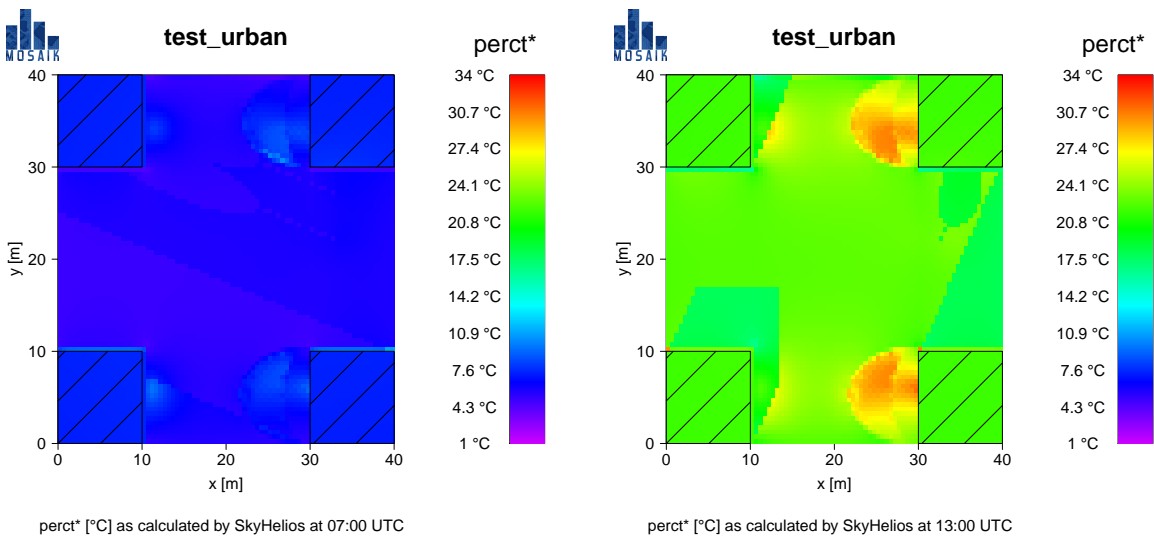

**Figure 7.** Perceived Temperature (PT, perct* in palm) for the test case "test_urban" at 07:00 UTC (left) and 13:00 UTC (right) on a 6[th] of March as calculated by the SkyHelios model.

show overall cooler conditions compared to the results by PALM (compare to Figures 4 and 5).

Comparing the results for 07:00 UTC on a 6[th] of March (Figures 4 and 7 (left)) the SkyHelios results generally are looking more homogeneous. This can be explained by air temperature and air humidity are considered static throughout the model domain in this comparison. Also the diagnostic wind model in SkyHelios generates more homogeneous wind fields in the absence of near-by obstacles. However, the results for PT are not only more homogeneous, but also significantly lower as calculated by SkyHelios than those by PALM. PT after SkyHelios ranges from 5.2 °C in the shade to a maximum of 11.4 °C in the sun in areas with very low wind speed (e.g. at the South-Western corner of the upper right building). This is way less than the PT calculated by PALM ranging from 10.2 to 15.6 °C in the sun and 8.0 to 11.1 °C in shaded areas. For the SkyHelios results, even the 3[rd] quantile of the PT results at 07:00 UTC of 7.9 °C is lower than the minimum value calculated by PALM.

A similar pattern can be found for the PT results at 13:00 UTC. Comparing Figures 4 and 7 (left) one can see once again that the SkyHelios results are more homogeneous for the reasons described above. However, the results calculated by SkyHelios are, again significantly lower than those by PALM. For the time of 13:00 UTC PT calculated by SkyHelios ranges from 16.6 °C in shaded areas to a maximum of 30.5 °C. The latter, however, is only reached in wind sheltered areas (West of the upper and lower right obstacle) that are exposed to direct radiation at the same time. Areas without the wind sheltering effect (e.g. in the central area of the domain) are significantly cooler (around 22.5 °C) even if they are exposed to direct radiation. PALM calculates way higher values of PT of 20.0 - 23.4 °C in the shade and 24.4 - 30.9 °C in the sun (see above).

Both, the differences at 07:00 as well as at 13:00 UTC can be explained by rather strong disagreement in the mean radiant temperature. While SkyHelios estimates mean radiant temperature of 4.7 °C in the shade to a maximum of 14.5 °C in the sun for the 07:00 UTC scenario, the same values are ranging from 11.4 °C to 35.4 °C in PALM. For the 13:00 UTC situation the

disagreement is little lower: While SkyHelios does calculate $T_{mrt}$ of 25.0 °C to 47.5 °C, PALM results in the range of 25.7 °C up to 51.7 °C.

## 3.2  Comparison to VDI versions and RayMan results

To get an insight on the precision of the results obtained from the biometeorology module, a direct comparison of results
by the thermal index programs published in the VDI guideline 3787 (VDI, 2008), as well as by the RayMan model was performed based on the same input data (please refer to Section 2.3 for details). The result for each index calculated by the biometeorology module fore a set of data was substracted by the respective VDI and RayMan version. An overview over the deviations is provided by Table 6.

**Table 6.** Statistical overview over the comparison of the results generated by the module for the thermal indices PT, PET and UTCI to those by the respective versions published in VDI guideline 3787 (VDI, 2008) and by the RayMan model.

|          | VDI | | | RayMan | | |
|----------|--------|--------|--------|--------|--------|--------|
|          | PT | PET | UTCI | PT | PET | UTCI |
| Min.     | -2.094 | -0.037 | 0.000 | -2.106 | -0.418 | -0.070 |
| 1st Qu.  | -0.003 | -0.004 | 0.000 | -0.044 | 0.019 | -0.020 |
| Median   | -0.001 | 0.003 | 0.000 | -0.009 | 0.052 | 0.000 |
| Mean     | -0.002 | 0.004 | 0.000 | -0.058 | 0.054 | 0.000 |
| 3rd Qu.  | 0.000 | 0.011 | 0.000 | 0.019 | 0.086 | 0.030 |
| Max.     | 1.356 | 0.083 | 0.000 | 2.683 | 0.488 | 0.070 |
| NA's     | 17073 | 17073 | 17073 | 17073 | 17073 | 211265 |

The comparison between the results for PT calculated by the biometeorology module and the VDI version reveils some
deviation of up to 2.094 K in rare cases (deviation of 0.1 K or more in 0.027 % of all cases tested in this study). The average deviations are found to be very low (0.002 K).

For the index PET the deviation between the results by the biometeorology module and the VDI version of the index is slightly higher in average (0.004 K) but does never reach a relevant level of 0.1 K (maximum of 0.083 K). Small deviations are to be expected due to rounding errors in the iterative PET calculations.

For UTCI no deviation can be found between the results generated by the biometeorology module and the VDI version of the index at all. This can be explained through UTCI is determined by the regression equation in both cases an, thus, is the least complex index in the comparison (no iterations).

The deviations to the results of the RayMan model are slightly higher for all indices. For PT, the deviation is up to 2.683 K, for PET the maximum deviation is is 0.488 K while there is only a slight deviation of up to 0.07 K for the index UTCI. The higher
deviations, however, can easily be explained by RayMan running on lower precision and rounding results to 0.1 K.

## 4  Discussion and Conclusions

The implementation of the thermal indices PT, UTCI and PET as a part of the newly developed biometeorology module does allow for a quantitative assessment of thermal comfort and thermal stress (e.g. Staiger et al., 2019) using the model PALM-4U (Maronga et al., 2019b). Results show that the human thermal comfort part of the biometeorology module can generate reliable
and plausible results for either of the indices in grid resolution for the vertical cell layer closest to 1.1 m above ground level. In the current version the most important indices for Germany are included. However, the module is open source and can easily be extended by the users favorite thermal index, e.g. the COMFA model (Brown and Gillespie, 1986).

The results presented in this study might seem quite high for the date of the case study, the 6[th] of March. However, with an air temperature ranging from 3.9 °C shortly after midnight to 23.9 °C in the afternoon the values are to be expected in this
region. Another reason for the hot conditions is the large radiational gain generated by the "clear-sky" scheme, that causes the mean radiant temperature to rise from -0.1 °C prior to sunrise to a maximum of 52.0 °C in the early afternoon. Furthermore considering the overall low wind speed, hot conditions as presented here are to be expected (e.g. Fröhlich et al., 2019).

Comparing the results calculated by the biometeorology module to those calculated by the SkyHelios model, the ones by Sky-Helios appear to be significantly lower. This is, as described above, mostly due to differences in the mean radiant temperature.
Also wind speed calculated by SkyHelios for an incident surface wind speed of 1.0 m/s from 270 ° is higher (around 0.1 m/s to 0.9 m/s) than the wind speed calculated by PALM (less than 0.1 m/s to 0.5 m/s at 07:00 UTC).

Both issues might be arising from the grid resolution used in the test calculations. With a grid resolution of 2.0 m on 2.0 m on 2.0 m the grid used for the PALM run is rather coarse. While this is required to keep the computational effort in reasonable scale for a complex model like PALM (Maronga et al., 2015) it decreases precision of the results (Fröhlich and Matzarakis,
2018). This definitely holds for the radiation calculations where the rather coarse obstacles throw stair-like shadows (Fröhlich and Matzarakis, 2018), but also for wind speed in the target height of 1.1 m. As 1.1 m is within the lowest possible layer of cells ground friction might be overestimated in the wind input to the biometeorology module. The SkyHelios model, in contrast, does perform radiation calculations in a vector-based model domain while the lower computational effort allows for higher target resolutions (Fröhlich and Matzarakis, 2018). To minimize the negative effects of the rasterized calculations in PALM, a
high resolution of e.g. 1 m on 1 m horizontally, as well as even higher vertical resolutions (e.g. with telescoping and nesting as proposed by Hellsten et al., 2018) is recommended by the authors.

The quality of the output of the biometeorology module is directly dependend on the input provided by PALM. Any uncertainties in the input values will be present in the thermal indices and therefore must be considered. PALM is a widely applied model, that has been used for various studies (e.g. Kanani et al., 2014; Kanani-Sühring and Raasch, 2015; Gronemeier et al.,
2017; Wang et al., 2017). Major imprecision therefore is unlikely. While the complexity of the model makes it difficult to assess the precision of the model as a whole, many modules were tested extensively in the past.

Most studies published about PALM are concentrating on wind speed and turbulence (e.g. Raasch and Schröter, 2001; Maronga et al., 2015). However, there is also some studies available on other parts of the model. The RRTM radiation scheme was described and assessed by Clough et al. (2005). Also the urban surface model was evaluated by Resler et al. (2017). They found

a slight overestimation in air temperature of 2 °C in the morning, but overall good agreement to their measurements as well as a "reasonably well" reproduction of the diurnal cycle (Resler et al., 2017). Resler et al. (2017) also compared building wall and street surface temperatures to measurements and found generally good agreement.

Considering the same input to the biometeorology module in terms of air temperature, moisture, wind velocity and mean radiant temperature, the output for PT, UTCI and PET does agree very well (considering the usual rounding effects) to reference calculations by the VDI version of the respective index as well as to results by the RayMan model (Matzarakis et al., 2007, 2010).

The new functionality implemented in the biometeorology module is intended to facilitate the consideration of several aspects of human thermal comfort and stress for various applications and user groups. This allows for the replacement of older and potentially less comprehensive models and methods not only in biometeorological research applications (e.g. Reis and Lopes, 2019; Nouri et al., 2018). It can be used by architects and municipalities to analyze the effect o their design on human thermal perception and health (e.g. Conti et al., 2005; Lin et al., 2010b; Fröhlich and Matzarakis, 2013) to improve their concepts e.g. fighting the local effect of global climate change or the urban heat island (Reis and Lopes, 2019).

*Code availability.* The specific version of PALM applied is provided in the folder SOURCE of Fröhlich (2019). In general, the PALM model system is free software. It can be redistributed and/or modified under the terms of the GNU General Public License (v3). We kindly request that you cite PALM in all your publications. It is available online as described in the PALM installation instructions: https://palm.muk. uni-hannover.de/trac/wiki/doc/install.

*Data availability.* The modified "test_urban" input dataset along with the results and the respective model source is available online along with Fröhlich (2019). It is a modification of the generic PALM test–case "test_urban" provided at https://palm.muk.uni-hannover.de/mosaik/ wiki/internal/testing (last access on 2019-06-19).

*Author contributions.* conceptualization, Dominik Fröhlich; methodology, Dominik Fröhlich and Andreas Matzarakis; software, Dominik Fröhlich and Andreas Matzarakis; validation, Dominik Fröhlich and Andreas Matzarakis; formal analysis, Dominik Fröhlich and Andreas Matzarakis; investigation, Dominik Fröhlich and Andreas Matzarakis; resources, Andreas Matzarakis, Dominik Fröhlich; data curation, Dominik Fröhlich; writing—original draft preparation, Dominik Fröhlich and Andreas Matzarakis; writing—review and editing, Andreas Matzarakis, Dominik Fröhlich; visualization, Dominik Fröhlich; supervision, Andreas Matzarakis; project administration, Andreas Matzarakis; funding acquisition, Andreas Matzarakis

*Competing interests.* The authors declare no competing interests.

*Acknowledgements.* This study is part of the MOSAIK project (https://palm.muk.uni-hannover.de/mosaik, last access May 11, 2020), a part of the $[UC]^2$ programme (http://www.uc2-program.org/index.php/en?page=structure_partner&lan=en, last access May 11, 2020), and is funded by the German Federal Ministry of Education and Research (BMBF).

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
