# Peer review of "Calculating human thermal comfort and thermal stress in the PALM model system 6.0"

_Geoscientific Model Development, 2019_

## Referee Comment (RC1) · Anonymous Referee #1 · 17 Oct 2019

This paper describes the development of the biometeorological module within the PALM modelling system. This new module will provide multiple new and existing applications within the outdoor urban thermal comfort research area. However, this manuscript requires extensive improvements and clarifications before I can recommend it to be accepted in GMD. I have listed my points of concern below in particular order.

1. The description on how mean radiant temperature (Tmrt) is derived from each voxel in PALM4U is not described. As this is one of the essential parameters when estimating outdoor thermal comfort this needs to be addressed and discussed. Also, as

[Figure]

in the results section, Tmrt derived from PALM4U deviates considerably from Rayman/SkyHelios as is used for comparison.

2. The other meteorological parameters needed for the thermal comfort indices (air temperature, humidity, wind speed) should also be described in detailed, and especially how they are used in the biomet module (e.g. what height are they derived from etc.). One follow-up question on this matter is based on a resolution in PALM test case of 2 meters; is the derived values coming from the center of a voxel (i.e. 1 meter height) of the corners (2 meter height)? Please clarify and discuss implications.

3. The authors state that UTCI is not suitable in some cases as the narrow range of valid input values are problematic. I assume this is mainly based on wind speed which has a lower limit of 0.5 m/s. But then comes the question how wind speed is recalculated up to the appropriate height which the regression model for UTCI is based upon, namely 10 meters above ground? Please clarify and justify the treatment of wind speed for UTCI. Furthermore, a "workaround" is mentioned (page 5, line 9). What workaround is this (never heard about this)?

4. iPT included in the methods section is not represented in the results and discussions and conclusions sections. Please insert examples of this or remove from methods or explain why this is not shown.

5. The input parameters in test_urban_p3d used (page 9 and 10) need further explanations. Please include.

6. The test runs are only presented for two occasions (0700 and 1300). What is interesting here is both the spatial variations as currently presented but also the temporal development of the input parameters as well as the resulting thermal indices. The model should be runned for a whole 24 h period so that the temporal variation can be examined and discussed. It is only then that solid conclusions and comparisons with other models can be made.

7. Page 10, line 23. What program is this referring to? Give more details.

8. Page 10, line 26. I totally disagree that figures 4-6 reveals that the cycle of thermal conditions is well reproduced. This is just 2 points in time and such conclusions cannot be drawn. See bullet point 6.

9. What wind direction and wind speed are used for the runs presented? This makes it hard to interpret the results. I cannot find this information anywhere in the text.

10. Where is the test-case located, Hannover?

11. The results when comparing SkyHelios and PALM4U outputs are confusing. Are the authors stating that the SkyHelios model and Rayman produce accurate results for the input parameters needed (e.g. Tmrt and wind) as well as the calculated thermal indices? This need to be discussed as providing Tmrt and wind are not trivial tasks and has been shown to give large deviations between models and field observations. Please elaborate and possible also include other models for comparison (Envi-met, SOLWEIG etc.) as well as field observations.

12. Why not include the COMFA model for deriving thermal comfort? This is a widely used index outside of Germany and could results in that more people will make use of this biomet module.

13. The clear-sky radiation in PALM seems to have a large effect on e.g. air temperature and Tmrt. Does this includes anisotropic sky irradiance and what levels of direct solar radiation is this clear-sky condition producing as input for the comfort calculations?

14. The PALM4U model results regarding the input parameters needs to be addressed, e.g. the large difference of air temperature (page 12, line 5-7) should be considered as unreasonable. Please discuss.

15. When comparing SkyHelios with PALM4U, please provide difference maps.

16. Section 3.2 is trivial and should be removed. That result can briefly be mentioned in the beginning of the results. However, it should also be stated that this comparison is made above roof level and is not representing a real street-level case which would probably change the result as surface temperatures etc. should be taken into consideration.

17. Matzarakis is present in more than 50% of the references which result in a way too high level self-referencing. This could undermine this work and not give it the high scientific quality that this work should have. Why not replace some of the references with work from other research groups e.g. Israel, Arizona, Gothenburg, Meinz etc.
* * *

---

## Referee Comment (RC2) · Anonymous Referee #2 · 26 Oct 2019

General remarks:

The article describes the implementation of PT, UTCI and PET in the biometeorological module of PALM-4U as well as the development of a new multi-agent tool. Results for PT and UTCI are shown and the results of PT are compared with the results of the model SkyHelios.

General remarks:

I am missing the precise description on how the agents who travel autonomously make their decisions. This is an important and interesting point in my opinion. The comparison with SkyHelios (but also between PT and UTCI) and its discussion needs a better

support by the presentation of additional parameters in Figures with discrete color bars (see also specific remark 10) in my opinion. Anonymous Referee#1 has already made many interesting points. I would be interested in the response of the authors.

Specific remarks:

1. P1, ln 13,14: Please rephrase

2. Figure 3: Add a north arrow please

3. P10, ln 26: The diurnal cycle is not shown, only two instances – maybe this should be pointed out.

4. Results: Please make it clearer whether you are referring to iPT. This new term was broadly discussed in the methods but in the Results only PT described.

5. Why are no PET maps shown?

6. Figure 4-7: Please include a North arrows. Due to the wide range of values a discrete color bar is absolutely necessary in my opinion.

7. P12, ln 10: I would rather use "wider range" than "a rather wide range". From slightly warm to warm is not really a big range in thermal sensation in my perception.

8. P13 ln 1: Are these now the PT values or UTCI. What are the UTCI values then?

9. P13, ln 4: I don't really see that why Figure 6 has more homogeneous areas than Figure 4. Could you mark them?

10. P16, ln5/5: I am not convinced, that a difference of 10 °C PT is a reliable and plausible result. I think it needs more additional analysis and maps of MRT and wind speed to better understand what causes these differences. Was MRT or wind speed validated for PALM or SkyHelios? Could you include this information?

Technical corrections:

P3, ln 6: for (not fpr)

P3, ln 18: Write the full name of the meteorological variable. If you decide on italics, please be consistent throughout the manuscript when you use Ta, VP, TMRT.

P3, ln 20: This is the first occurrence but it applies though out the manuscript: Be consistent either use Figure 1, Section 1, ... or fig. 1., sec. 1, tab.1, ... )

P4, ln 1: missing and between the citations

P4, ln 18/19: There seems to be something wrong in this sentence.

P5, ln 6: I wouldn't call it a narrow, but limited range. In relative humidity it is not even limited.

P7, ln 10: lagrangian (not langranian)

P9, Figure 3: on 6th of March (not on a 6th of March).

P9, ln 2,3: shouldn't it be "...there are buildings with different heights ..."?

P9, ln 6,7: Is the formatting of the doi correct?

P10, ln 2: 10 minutes (not 10 Minutes)

P10, ln 27: Only reference to Figure 4, not 5

P11, ln ? : Make "figure 4" consistent with the other references.

P12, ln 12: Tmrt or mTmrt?

P13: Figure 6: Missing blank space between Thermal and Climate

P13, ln 4: ...Figure 6 one can see... (not "on can see")

P14, ln 5: minimum value (not "minim")

P14, ln 5: one can see ... (not "on can see...")

P15, ln 14: and thus?

---

## Author Comment (AC1) · 4 Nov 2019

Dear Referee #1:

The authors want to express their gratitude for your detailed and valuable comments. Please find your comments addressed in detail below.

1: The mean radiant temperature is mostly determined in the radiation module, not the biometeorology module. This is while it was considered out of scope for this manuscript. The authors, however, do see the relevance of the parameter. A description therefore will be added to the methods.

2: All input parameters are determined for the center of the voxel, that is closest to 1.1m above ground level. The cell level is determined in the initialization of the module.

3: The limitation of the regression mostly arises from spatially resolved wind speed being lower than 0.5 m/s in 10m height. The wind speed in 10 m above ground level is determined from extrapolating the wind speed at the cell center level closest to 1.1m above ground level (se above) by applying the wind profile from the original UTCI determination (Havenith, UTCI clothing). The workarounds mentioned in the manuscript are those published by Broede et al. 2012.

4: This arises from iPT is integrated into the BiometMod, but not correctly called from the agent module in version 6 that is the basis of this manuscript. We will check if it is suitable to show results anyway or remove the index from the methods.

5: The parameters listed there are modifications to the original generic test dataset provided at https://palm.muk.uni-hannover.de/mosaik/wiki/internal/testing. What parameter is unclear from your point of view?

6: For keeping the manuscript at a reasonable size only two exampleas are presented here.     However, the entire dataset with input and output is published along with the manuscript and can be found (as stated in the manuscript) at https://zenodo.org/record/3433720.

7: Page 10 line 23 reads "model. The general statistics of the dataset is provided by Table 4. The output generated by the biometeorology module was". Is this the line you are referring to?

8: Page 10 line 26 was intended as some introductory sentence rather than as a statement.  The authors agree that this could be misleading.  It will be modified to e.g. "Results for the test case (e.g. Figures 4 - 6) show the changing thermal conditions over the day."

9: Incident wind is from 270° with 1.0 m/s. Added to the respective figures. Thanks for

the hint!

10: Yes. Also added to the methods.

11: This manuscript is about the calculation of thermal indices within the biomet module in PALM. The assessment of the input parameters provided by PALM and its other modules is not in scope of the manuscript (and can hardly be done by the authors as those parts are by different developers). Neither Envi-met nor SOLVEIG can calculate the thermal indices PT, PET and UTCI what makes them not relevant for this manuscript.

12: The module was developed in the course of a German project. Therefore the most important indices for Germany are included. However, the module is open source and can easily be extended by the users favorite thermal index.

13: For information about the radiation schemes in PALM please see the palm documentation at https://palm.muk.uni-hannover.de/trac/wiki/doc/tec/radiation.

14: This manuscript can not assess the results of the PALM model core or the radiation module that are used as input parameters. This is up to the respective module authors. The manuscript is about the thermal indices only.

15: For the models are of different types (SkyHelios is a diagnostic model) and resolutions, difference maps can hardly be produced here. They will also hardly help as the maps are shown to compare the spatial patterns and not individual values.

16: Section 3.2 is definitely neither trivial nor can it be removed. It is the closest-possible to a direct comparison that is required to assess the quality of the thermal indices calculated. The location the input values were recorded is irrelevant as they are exactly the same values for the biometeorology module, the RayMan module and the VDI versions of the indices. Also Tmrt is the very same row of data for all three models.

17: Matzarakis is not a (co-) author in 28 of 55 references (50.9%). The authors

however agree that this can be reduced and will remove some of the references.

**[GMDD](https://doi.org)**

Interactive
comment

---

## Author Comment (AC2) · 4 Nov 2019

Dear Anonymous Referee #2:

The authors want to thank you for your detailed analysis of their work and your efforts in improving the manuscript. Please find your comments addressed in detail below.

General: - The agent module along with the decisions the agents are making can unfortunately not be described in this manuscript as this is hardly in scope. However, there is a vast documentation of the agent module available online. Please have a look at https://palm.muk.uni-hannover.de/trac/wiki/doc/tec/mas. - The authors see the point

in providing maps showing the input parameters ass well. However, this would result in alt least four input parameter maps (air temperature, vapor pressure, wind speed and mean radiant temperature) in addition to the result maps what can hardly be presented here. However, there might be a way to submit this as a kind of an attachment. The data can already be analyzed as it is provided along with the entire test dataset at https://zenodo.org/record/3433720.

Specific: 1. Modified to "Results show deviations below the relevant precision of 0.1∼K for PET and UTCI and some deviations of up to 2.683∼K for PT caused by repeated unfavorable rounding in very rare cases (0.027∼\%)." Better? 2. In geography north arrows are neglected of the North is right up what is the case here. 3. Yes, this was already noted by anonymous referee #1. The sentence was therefore modified to "Results for the test case (e.g. Figures 4 - 6) show the changing thermal conditions over the day." 4. Unfortunately there is not results for iPT in the current version. This is due to issues with the agent module in the PALM version 6.0 that is lacking the interface to the biometeorology module. Requests to the module maintainer is still pending. If this is not addressed, iPT has to be removed entirely from the manuscript. 5. The maps presented in the manuscript are to show the general patterns of the thermal indices. The ones of PET therefore agree rather well to the ones for PT. They are therefore neglected to keep the manuscript as short as possible. 6. See specific comment #3. In contrary, I think a discrete color ramp would cause all of the map being in the same class of values obfuscating any details. 7. Agreed and replaced. 8. They are both UTCI. Added to the manuscript to avoid misunderstandings. 9. This is indicated in the next sentence, but I admit it is hard to see in the graph. 10. The statement needs to be interpreted in connection to the direct comparison with fixed input as well. Considering the input, the output of the module is perfectly plausible. Additional maps would be beneficial but need a lot of space (and there would be a lot of them if input and full diurnal cycles are considered). The authors however will check if additional maps can be uploaded as supplementary material.

Thanks for the many technical comments. All fixed.

---

## Referee Comment (RC3) · Anonymous Referee #1 · 7 Nov 2019

I will submit more detailed comments when I have read the updated manuscript.

In the mean time, here are answers (comments) to some of the authors questions (comments):

3. Why not retreive wind speed from 10 meters above ground from the output of palm instead of recalcualting is separately? I you do decide to do this, plesea justify and present the differences and methodology used for the recalculation procedure.

5. No parameters are unclear but I would argue that a model description paper like this should explain the items included and not refer back to any test dataset etc. If you choose to include this stearing parameters, please give explanations of the same.

7. In my version line 23 on page 10 is "...then compared to the output by the programs in the attachment to the VDI guideline 3787...". The VDI guideline programvis confusing to me. Is this a software of is it just information text or something else? Please clarify.

17. Sorry, my misstake. It should be 27 of 55 references (49.1%). Hence, my argument of self-referencing still holds.

---

## Short Comment (SC1) · 25 Nov 2019

This is an executive editor comment highlighting the ways in which this manuscript is not currently compliant with GMD policy on code and data availability. In this case, the code and data availability section is completely inadequate and needs to be substantially improved before a revised manuscript can be accepted.

1. Code availability. The code availability is simply given as a project SVN repository. this neither identifies the exact version employed, nor meets GMD requirements for persistence and non-revocability. Since PALM is GPL, there is nothing

preventing a persistent, public archive of the exact version employed in this work being created on a suitable archive such as Zenodo. This needs to happen.

2. The input data, run scripts and analysis scripts used to conduct the tests are not cited from this section at all. There are some Zenodo DOIs in the main text body, but GMD policy requires this information to be locatable by going to the code and data availability section. Please note also that placing Zenodo DOIs in any part of the text is not good practice. Instead, the correct bibliography entry should be retrieved from the Zenodo entry itself and included in the bibliography. This should be cited from the text.

Further details on code and data availability requirements are in the GMD model code and data policy: https://www.geoscientific-model-development.net/about/code_and_data_policy.html. The reasons for the policy and more detail are provided in this editorial: https://doi.org/10.5194/gmd-12-2215-2019

---

## Author Comment (AC3) · 9 Dec 2019

Dear Anonymous Referee #1,

thanks for your valuable comments and your efforts in improving the manuscript. We addressed your comments individually below.

3. Why not retreive wind speed from 10 meters above ground from the output of palm instead of recalcualting is separately? I you do decide to do this, plesea justify and present the differences and methodology used for the recalculation procedure.

The wind speed actually used for UTCI is 1.1 m as well. It is determined from the

10 m wind speed by a profile published in Havenith et al. 2012. For this was also applied in the generic runs that are the source to the regression equation wind input must be provided for 10m above ground level. The module applies the very same profile to actually directly use the wind speed at the desired altitude with minimal error. Furthermore the model allows for rather complex environments. UTCI might therefore be desired for places where wind speed is available at (approximately) 1.1m, but not at 10 ̆am above ground level (e.g. below some obstacle).

5. No parameters are unclear but I would argue that a model description paper like this should explain the items included and not refer back to any test dataset etc. If you choose to include this stearing parameters, please give explanations of the same.

The list was removed entirely as the whole input is provided anyway and it was only relevant for the difference to the generic scenario, that is not part of the manuscript any more.

7. In my version line 23 on page 10 is "...then compared to the output by the programs in the attachment to the VDI guideline 3787...". The VDI guideline programvis confusing to me. Is this a software of is it just information text or something else? Please clarify.

A reference version of the three thermal indices implemented in the biometeorology module is provided by a compact disc shipped along with the VDI guideline including both compiled versions as well as source code.

17. Sorry, my misstake. It should be 27 of 55 references (49.1%). Hence, my argument of self-referencing still holds.

Agreed anyways. Self-referencing was reduced.

---

## Author Comment (AC4) · 9 Dec 2019

Dear David Ham,

thank you very much for the hint!

1. Code availability. The code availability is simply given as a project SVN repository. this neither identifies the exact version employed, nor meets GMD requirements for persistence and nonevocability. Since PALM is GPL, there is nothing preventing a persistent, public archive of the exact version employed in this work being created on a suitable archive such as Zenodo. This needs to happen.

A folder SOURCE was added to the dataset at zenodo holding the specific version of PALM.

2. The input data, run scripts and analysis scripts used to conduct the tests are not cited from this section at all. There are some Zenodo DOIs in the main text body, but GMD policy requires this information to be locatable by going to the code and data availability section. Please note also that placing Zenodo DOIs in any part of the text is not good practice. Instead, the correct bibliography entry should be retrieved from the Zenodo entry itself and included in the bibliography. This should be cited from the text. Further details on code and data availability requirements are in the GMD model code and data policy: https://www.geoscientific-model-development.net/about/code_ and_data_policy.html. The reasons for the policy and more detail are provided in this editorial: https://doi.org/10.5194/gmd-12-2215-2019

The zenodo references were added to the references section. The doi and url in the text have been replaced by the citation.

---

## Author Response (AR1)

Dear Editors,

there is no comments to answer in this round. Please see the public discussion and our responses to the anonymous referees as well as to David Ham.

Please further find an annotated version showing the differences in the manuscript attached below.

[revised manuscript text omitted]

---

## Author Response (AR2)

Dear editor,

thank you for your efforts in improving the manuscript! Please find the answers to the specific comments in detail below.

Comments related to Referee #1
1. You have added some description on the mean radiant temperature but this new text describes mainly the radiative models from where mrt can be calculated but not the details referee #1 was asking. Due to the importance of this variable, even though it is calculated in the radiation module, you should provide more detailed description from each voxel as the referee was requesting.

For a more detailed description of the mean radiant temperature calculations in PALM, please see the Maronga et al. 2019a (Overview of the PALM model system 6.0) section 3.6. If included in this paper would be repetition while is does not belong to the biometeorology module.

2. The comments on better description of meteorological parameters in the manuscript has not addressed. You only mention the height in the referee response but this is not enough to answer the referee #1 comment.

The meteorological parameters used for the calculations with the biometeorology module are provided by the PALM-output directly. The biometeorology module does only use, but not modify any of the input parameters (air temperature, relative humidity, wind speed and mean radiant temperature).

3. The wind speed calculation is not addressed in the manuscript itself and thus more information should be added.

Wind speed is provided by the PALM model core and only used for input to drive the module. For details please see the PALM model description papers (e.g. Maronga et al. 2015, Maronga et al. 2019a).

4. What happened to the description of iPT? This was left open and its not clear how the referee comment was answered

iPT was removed in the manuscript for the agent module is still not fully available. If desired, however, it can be added again at any time.

6. It would be possible to include some temporal plots in addition to the spatial "snapshots". You could add a temporal plot with e.g. four subplots on certain points (wall, floor,…) from PALM-4U and from Rayman. This would be important evaluation on showing how the biomet module behaves on temporal scale.

RayMan is a diagnostic model type that can not be compared that easily and appropriately to a prognostic model like PALM. A temporal comparison like proposed above would rather give an insight on the model core's precision providing the input data to the biometeorology module than the module itself.
A temporal comparison to RayMan would, furthermore, lead to the very same results as the biomet module for RayMan would need to be driven with the same input data to test the accuracy of the biometeorology module rather than the PALM model core.

7. Please, add the information also to the manuscript as this might be unclear for other readers also.

Has been added to the manuscript (Section 3, Paragraph 1).

8. It still reads in the manuscript over the day which is not true as currently only two points are shown. I suggest you actually add temporal plot comparing Rayman and PALM 4U as suggested above in point 6.

The sentence has been replaced (Section 3, Paragraph 1 and 2).

9. The meteorological conditions should be added to the methods under Test case also.

A table showing the initial conditions has been added to the manuscript (Section 2.2, Table 4).

11 Discussion concerning uncertainty in the input parameters should be added to the manuscript as suggested by the referee

The uncertainties of the input parameters of course directly do affect the output of the biometeorology module. The accuracy of the input conditions can not be assessed by the authors of the biometeorology module. Please see Maronga et al. 2019a.

12 It would be good to include this information to the manuscript as requested by the referee

Added to the manuscript (Section 4, Paragraph 2).

16 You have highlighted text in Section 3.2 but it is not clear what has changed. It would be good to add lines that have changed to the referee responses as now it is difficult to follow what changes exactly has been made and to what referee comments do the changed concern.

This might be related to pdfdiff missed a page break. Please ignore.

17. Not clear what references have been removed and also no references to other studies as suggested by the referee have ben added.

The following references have been added to or removed from the manuscript since the initial submission.
Added references:
- Brown, R. D. and Gillespie, T. J.: Estimating outdoor thermal comfort using a cylindrical radiation thermometer and an energy budget model, International Journal of Biometeorology, 30, 43–52, https://doi.org/10.1007/BF02192058, 1986.
- Fröhlich: Modified "test_urban" dataset for thermal comfort in PALM, https://doi.org/10.5281/zenodo.3567814, https://doi.org/10.5281/zenodo.3567814, 2019.
- Maronga, B., Banzhaf, S., Burmeister, C., Esch, T., Forkel, R., Fröhlich, D., Fuka, V., Gehrke, K. F., Geletič, J., Giersch, S., Gronemeier, T., Groß, G., Heldens, W., Hellsten, A., Hoffmann, F., Inagaki, A., Kadasch, E., Kanani-Sühring, F., Ketelsen, K., Khan, B. A., Knigge, C., Knoop, H., Krč, P., Kurppa, M., Maamari, H., Matzarakis, A., Mauder, M., Pallasch, M., Pavlik, D., Pfafferott, J., Resler, J., Rissmann, S., Russo, E., Salim, M., Schrempf, M., Schwenkel, J., Seckmeyer, G., Schubert, S., Sühring, M., von Tils, R., Vollmer, L., Ward, S., 18Witha, B., Wurps, H., Zeidler, J., and Raasch, S.: Overview of the PALM model system 6.0, https://doi.org/10.5194/gmd-2019-103, https://www.geosci-model-dev-discuss.net/gmd-2019-103/, 2019a.
- Mlawer, E., Taubman, S., Brown, P., Iacono, M., and Clough, S.: RRTM, a validated correlated-k model for the longwave, J. Geophys. Res., 16, 663–682, 1997.

- Pincus, R., Barker, H. W., and Morcrette, J.-J.: A fast, flexible, approximate technique for computing radiative transfer in inhomogeneous cloud fields: FAST, FLEXIBLE, APPROXIMATE RADIATIVE TRANSFER, Journal of Geophysical Research: Atmospheres, 108, https://doi.org/10.1029/2002JD003322, http://doi.wiley.com/10.1029/2002JD003322, 2003.
- Thorsson, S., Lindberg, F., Eliasson, I., and Holmer, B.: Different methods for estimating the mean radiant temperature in an outdoor urban setting, International Journal of Climatology, 27, 1983–1993, https://doi.org/10.1002/joc.1537, wOS:000251432100012, 2007.

Removed references:

- Herrmann, J. and Matzarakis, A.: Mean radiant temperature in idealized urban canyons – Examples from Freiburg, Germany, International Journal of Biometeorology, pp. 199–203, 2012.
- Lin, T.-P., Tsai, K.-T., Liao, C.-C., and Huang, Y.-C.: Effects of thermal comfort and adaptation on park attendance regarding different shading levels and activity types, Building and Environment, 59, 599–611, https://doi.org/10.1016/j.buildenv.2012.10.005, http://www.sciencedirect.com/science/article/pii/S0360132312002703, 2013.
- Matzarakis, A. and Mayer, H.: Mapping of urban air paths for planning in Munich, In: Planning Applications of Urban and Building Climatology. Wissenschaftliche Berichte des Instituts für Meteorologie und Klimaforschung. Universität Karlsruhe, 16, 1992.
- Matzarakis, A., Mayer, H., and Iziomon, M. G.: Applications of a universal thermal index: physiological equivalent temperature, International Journal of Biometeorology, 43, 76–84, https://doi.org/10.1007/s004840050119, wOS:000083502400004, 1999.
- Matzarakis, A., Röckle, R., Richter, C.-J., Höfl, H.-C., Steinicke, W., Streifeneder, M., and Mayer, H.: Planungsrelevante Bewertung des Stadtklimas – Am Beispiel von Freiburg im Breisgau, Gefahrstoffe – Reinhaltung der Luft, pp. 334–340, 2008.
- Matzarakis, A., Muthers, S., and Koch, E.: Human-biometeorological evaluation of summer mortality in Vienna, Theoretical and Applied Climatology, pp. 1–10, 2011.

Comments related to Referee #2

To me it seems that the agent model is highly relevant for iPT and thus more detailed description should be added to the current manuscript. This does not need to be long but more details as suggested by referee #2 should be added.

iPT was removed from the manuscript entirely for the agent model is still not fully available.

I do not see whether such plots (input spatial plots) were produced to attachments. If not the authors should provide them as suggested. These can be added to the main text of to the attachments.

A temporal comparison like proposed above would rather give an insight on the model core's precision providing the input data to the biometeorology module than the module itself.
A temporal comparison to RayMan would, furthermore, lead to the very same results as the biomet module for RayMan would need to be driven with the same input data to test the accuracy of the biometeorology module rather than the PALM model core.

Please add line numbers and pages where the specific changes have been made as now it is difficult to read the changes.

Line numbers are changing along with any modification of the manuscript and can therefore be misleading. We therefore highlighted the changes to the last uploaded version as demanded by the authors instructions.

[revised manuscript text omitted]